# A Monte Carlo Determination of Dose and Range Uncertainties for Preclinical Studies with a Proton Beam

**DOI:** 10.3390/cancers13081889

**Published:** 2021-04-15

**Authors:** Arthur Bongrand, Charbel Koumeir, Daphnée Villoing, Arnaud Guertin, Ferid Haddad, Vincent Métivier, Freddy Poirier, Vincent Potiron, Noël Servagent, Stéphane Supiot, Grégory Delpon, Sophie Chiavassa

**Affiliations:** 1Institut de Cancérologie de l’Ouest, 44800 Saint-Herblain, France; arthur.bongrand@ico.unicancer.fr (A.B.); Daphnee.Villoing@ico.unicancer.fr (D.V.); vincent.potiron@univ-nantes.fr (V.P.); stephane.supiot@ico.unicancer.fr (S.S.); Gregory.Delpon@ico.unicancer.fr (G.D.); 2GIP ARRONAX, 44800 Saint-Herblain, France; koumeir@subatech.in2p3.fr (C.K.); haddad@subatech.in2p3.fr (F.H.); poirier@arronax-nantes.fr (F.P.); 3Laboratoire SUBATECH, UMR 6457, CNRS IN2P3, IMT Atlantique, Université de Nantes, 44307 Nantes, France; Arnaud.Guertin@subatech.in2p3.fr (A.G.); Vincent.Metivier@subatech.in2p3.fr (V.M.); Noel.Servagent@subatech.in2p3.fr (N.S.)

**Keywords:** radiation dosimetry, Monte Carlo simulation, preclinical studies, proton therapy, dose calculation

## Abstract

**Simple Summary:**

If reference studies can be found on the uncertainties linked to the clinical context of proton therapy, they, although equally critical, are very patchy in a preclinical context, and are specific to each beam line. This work provides the community with a complete assessment of the sources of uncertainties for a preclinical proton beam line. This aims to ensure that, in this line, the biological results observed and the dose–response relationships are obtained without any bias. Despite being specific to a preclinical line, the results presented here can be transposed to other types of proton preclinical facilities, and thus allow us to effectively compare them to one another.

**Abstract:**

Proton therapy (PRT) is an irradiation technique that aims at limiting normal tissue damage while maintaining the tumor response. To study its specificities, the ARRONAX cyclotron is currently developing a preclinical structure compatible with biological experiments. A prerequisite is to identify and control uncertainties on the ARRONAX beamline, which can lead to significant biases in the observed biological results and dose–response relationships, as for any facility. This paper summarizes and quantifies the impact of uncertainty on proton range, absorbed dose, and dose homogeneity in a preclinical context of cell or small animal irradiation on the Bragg curve, using Monte Carlo simulations. All possible sources of uncertainty were investigated and discussed independently. Those with a significant impact were identified, and protocols were established to reduce their consequences. Overall, the uncertainties evaluated were similar to those from clinical practice and are considered compatible with the performance of radiobiological experiments, as well as the study of dose–response relationships on this proton beam. Another conclusion of this study is that Monte Carlo simulations can be used to help build preclinical lines in other setups.

## 1. Introduction

Radiation therapy consists in delivering ionizing radiation to the tumor while preserving surrounding normal tissues. Tumor resistance and the treatment toxicity to healthy organs often limit the treatment efficacy. Consequently, a compromise must be found between possible risks and benefits. The dose–response curves for tumor control and normal tissue complications help define a therapeutic window, to deliver a sufficient dose of radiation to the tumor with acceptable side effects [1]. In this attempt to limit toxicity, proton therapy is an option for some tumor locations, as this irradiation technique spares most of normal tissues. In contrast with photons, protons and light ions stop depositing energy at a given depth, with relatively low straggling, leading to a significant reduction in radiation dose to surrounding normal tissues. In addition, the proton beam delivery system allows high-precision radiation deposition in the three dimensions of the tumor [2]. 

ARRONAX is a cyclotron facility, which partially serves research. It hosts a multi-particle accelerator that can produce a broad range of radiations: protons (30–70 MeV), deuterons (15–35 MeV), and alpha particles (68 MeV) [3]. The characteristics of the ARRONAX proton beam are therefore adapted to preclinical research on cells and small animals. 

In clinical practice, an incorrect estimation of the proton range impacts the absorbed dose to the target and the organs at risk, which can have major consequences for the patients. In preclinical studies, the impact is lower. However, any error in the predicted trajectory will introduce a significant bias in the observed biological outcomes and the dose–response relationship. Therefore, the range of proton beams in biological samples needs to be planned as accurately as possible, in order to produce reliable data. In preclinical as in clinical practice, uncertainties depend on the approximations of the dose calculation and the irradiation setup. 

Furthermore, the ARRONAX environment was not directly intended for clinical proton therapy. Some uncertainties are therefore to be expected, due to the proton beam and its components. Given the specificity of each beam line, the American Association of Physics in Medicine recommends in its Report of Task Group 202 [4] to determine the uncertainties for each line and to report them to the community. If global uncertainty can be assessed with experimentation, only Monte Carlo simulation enables us to discriminate, quantify, and minimize individual uncertainty components [5]. 

In this study, we followed up on Paganetti’s work [6] and used Monte Carlo simulation to evaluate the impact of various sources of uncertainty on the proton range, the absorbed dose, and the dose homogeneity, in the context of preclinical irradiation of cells or small animals on the plateau. 

## 2. Materials and Methods

### 2.1. ARRONAX Facility 

ARRONAX is an isochronous cyclotron (IBA Cyclone 70XP) that provides bunches of protons interspaced by 32.84 ns (RF = 30.45 MHz). Proton beams can be produced from low (<1 pA) to high (up to 350 μA) intensities. Using a homemade pulsing device [7], it is possible to precisely set up the duration of the irradiation from a few microseconds to a few seconds, as well as the frequency rate of repetition. The ARRONAX cyclotron offers the possibility of delivering a given dose, ranging from a low dose rate (<1 mGy/s) with conventional irradiation to ultra-fast irradiation at a high dose rate (>1 mGy/s), allowing both conventional and very high dose rate (VHDR) experiments. 

Proton energy (68 MeV, range ≈ 3.5 g/cm^2^) was adapted for the irradiation of small biological samples, such as cells, zebrafish embryos, or small animals [8]. Examples of dose calculations computed on mouse cone beam computed tomography (CBCT) images for the ARRONAX proton beam are provided for lung, brain, and intestinal irradiation in the Bragg curve (Figure 1). The experimental setup is detailed in Figure 2 and Table 1. The collimator‘s diameter varied from 7 mm to 20 mm. For beam diameters greater or equal to 10 mm, a scatterer (thin tungsten foil) was also added to homogenize the beam with a very small impact on the beam energy. In our study, we simulated two types of targets: the biological sample and the mouse. The biological sample, such as cells or zebrafish embryos, is always placed in liquid during the irradiation. Therefore, it is assimilated to a water cylinder, 4 cm long, along the beam axis.

### 2.2. Monte Carlo Simulations

In his study of range uncertainties in clinical proton therapy [6], Paganetti concludes that for complex geometries, Monte Carlo radiation transport techniques might reduce the range uncertainty by several mm in comparison with analytical algorithms. Range uncertainties with Monte Carlo are shown to be ±0.1% for both complex and local lateral inhomogeneities, but −0.7% and ±2.5% with analytical algorithms, respectively. Hence, we opted for Monte Carlo simulations in the present study. 

Simulations were performed with the Monte Carlo code GATE v9.0 [9], which is based on Geant4 v10.06p01 [10]. The reference hadronic physics list recommended for proton therapy was applied: QGSP_BIC with electromagnetic option 4, EMZ [11]. The modeled proton source was elliptical (FWHM 2 and 3 mm) and placed in a vacuum, 10 µm before the Kapton window, in accordance with experimental observations. The cut production for secondary *e*−, *e*+, and γ particles was set to 10 mm in all geometries and 0.1 mm in the targets. The maximum step size was limited to 1 mm in all geometries and 100 μm in the targets. The number of initial particles was set up to 500 million, to keep statistical uncertainty below 2% for each simulation. The calculation grid size was 0.2 × 0.2 × 0.2 mm^3^. and dose-to-water was selected for the simulation.

### 2.3. Sources of Uncertainties

Sources of uncertainty, independent of or dependent on dose calculation, were considered separately. Sources of uncertainty independent of dose calculation are linked to the commissioning measurement, the beam (energy spectrum and beam component), and the target setup. Those dependent on dose calculation concern complex targets and are mainly due to computed tomography (CT): spatial resolution, materials, densities, and mean excitation energy (*I*-value) assignment. The range degradation due to complex and local lateral inhomogeneities is another source of uncertainty dependent on dose calculation, estimated at ±0.1% by Paganetti [6]. For dose calculations in biological samples mainly composed of water, the *I*-value of water is also a source of uncertainty. 

The estimated relative standard uncertainty for dose commissioning measurement was set to ±2.0%, according to the IAEA TRS-398 [12]. The other uncertainties were assessed by Monte Carlo simulation. 

#### 2.3.1. Beam Energy Spread

Similar to clinical systems, the energy of the ARRONAX proton beam come with an energy spread. This energy spread is a source of uncertainty for the proton range and the absorbed dose in the Bragg curve. The theoretical energy spread of the 68 MeV beam is ±0.68 MeV. In this study, however, we considered a mono-energetic beam of 68 MeV as a reference and a maximal energy spread of ±1 MeV. The evaluation was performed in a water target, with and without the tungsten foil.

#### 2.3.2. Beamline Components

Despite their small thicknesses, the beamline components described in Figure 2 and Table 1. reduce the beam mean energy by about 2% and increase its energy spread. Their composition and density are 99.9% guaranteed by the manufacturer, but their thickness is provided with an uncertainty of ±5%. We studied the impact of this range of thickness for each beamline component.

Moreover, the setup must be reinstalled at each experiment, leading to a variation in component positioning. We evaluated the impact of each element’s positioning in a range of ±1 cm.

As reported by Paganetti [6], the mean excitation energy is one of the main sources of uncertainty for the proton range determination. Elemental *I*-values were estimated experimentally by Seltzer and Berger [13]. These values were then adapted in 1984 in International Commission on Radiation Units and Measurements (ICRU) report 37 [14], and were taken over by ICRU report 49 [15] for proton therapy. For compounds, experimental determination is not possible, and the Bragg additivity rule (BAR) is applied. Geant4 established the Geant4 Material Database following these recommendations. *I*-values for the beam component evaluations were taken from this database, as well as the corresponding uncertainties when available. For unavailable uncertainties, ±6% was applied, according to Bär et al. [16]. Applied *I*-values and a range of uncertainties are given in Table 2 for each material.

The impact of beam components in term of thickness, position, and *I*-value was estimated in the water target, with and without the tungsten foil.

#### 2.3.3. Target Position in the Beam Axis

Biological samples, either cells, zebrafish embryos or mice, are placed on adapted supports with motorized axes in order to adjust the target position according to the beam entrance. For in vitro experiments, 2D sample targeting is accurate, repeatable and guaranteed by the use of radiochromic films, placed perpendicularly to the beam. For in vivo experiments, mice are anesthetized with isoflurane throughout the irradiation process and can be correctly positioned in the beamline on a specific support. In our configuration, the targeting of an organ or a tumor can only be adjusted using an offline imaging system with immobilization systems and external landmarks. 

The impact of uncertainty in the positioning of biological samples and mice in the beam direction was evaluated, with a maximal shift of ±1 cm. For mice, brain, lungs, and intestine localizations were studied. 

#### 2.3.4. CT Imaging and Calibration

Monte Carlo simulations rely upon information on density and material composition of each irradiated volume. When considering a complex target (patient or small animal), Hounsfield units (HU) of the target CT imaging must be converted into material/density data using stoichiometric approaches, based on CT scans of tissue phantom materials with known density and elemental composition. In this work, target imaging was performed with a CBCT device (XRAD225Cx, PXI, 40 kVp), and an adapted conversion method was applied [17] in order to automatically convert HU values into materials. This conversion is based on a proper stoichiometric calibration, which relies on the use of reference materials. 

In a clinical situation, considering more than 10 different tissues already leads to a range uncertainty of ±0.5% [6,18] caused by CT image noise. In this work, the materials are assigned according to a database of 125 tissues interpolated from ICRU reports 44 [19] and 46 [20]. Due to the acquisition technique, a higher noise level is expected in CBCT than in conventional CT images. In our case, the range of variation was estimated to ±40 HU [17]. The impact on absorbed dose, proton range, and homogeneity index (HI98) was estimated, artificially modifying CBCT images in the range of ±40 HU, considering brain, lungs, and intestinal localizations.

#### 2.3.5. CT Voxel Size

The accuracy of anatomical definition of complex targets, from either CT or CBCT images, depends on image grid size. Due to the partial volume effect, this spatial resolution can affect dose calculation, particularly in the case of high-density interfaces (lungs, bones). Our CBCT (XRAD225Cx, PXI, 40 kVp) resolution is adapted to small animal size, with a voxel size of 0.2 × 0.2 × 0.2 mm^3^. 

The impact of image resolution was evaluated for brain, lungs and intestine irradiations by degrading the grid size to 0.4 × 0.4 × 0.4 mm^3^ and 0.8 × 0.8 × 0.8 mm^3^ (Figure 3), while maintaining the calculation grid size at 0.2 × 0.2 × 0.2 mm^3^ to prevent the accumulation of sources of uncertainties.

#### 2.3.6. *I*-Values in Water and Tissues

A wide range of *I*-values for water is available in the literature, from 67.2 eV [21] to 80.8 eV [22]. In their report 90 [23], the ICRU recommends the use of 78 eV for the mean excitation energy of water with a range of uncertainty of ±2 eV. The impact of the *I*-value for the water target was evaluated in this range. 

For compounds like biological tissues, the BAR method is applied by default in GATE. Nevertheless, this method of calculation remains theoretical, and must be constrained as much as possible by experimental measurements. Although optimized *I*-values based on BAR for human tissues are available in the literature [16,21,24], Geant4’s recommended values are those from the ICRU [21]. To determine optimized *I*-values of the 125 tissues interpolated using the Noblet method [17], we used optimized *I*-values determined by Bourque et al. [25] for 34 tissues. *I*-values were then interpolated for the 91 remaining tissues. Regarding the uncertainties, the ICRU does not provide them for tissue *I*-values. 

We used the estimates from Bär et al. [16], and applied a maximum uncertainty of ±6% when evaluating the impact of the *I*-value for tissues.

### 2.4. Studied Parameters 

In this work, we evaluated the impact of each source of uncertainty for the absorbed dose in the plateau and on the proton range, defined as the distance between the entrance surface of the beam and the distal point of the 80% dose. In the context of irradiation in the plateau, the Bragg peak must be maintained outside the irradiated sample. For the mouse simulation, a water tank was placed behind the mouse to observe the variation in the Bragg peak position. Additionally, due to the increasing energy deposition before the Bragg peak, a modification of the proton range could affect the homogeneity in the target. 

To analyze and quantify the dose homogeneity in a target volume, clinicians use an objective tool, called the homogeneity index (HI). The HI value indicates the ratio between the maximum and minimum dose in a target volume. The ideal HI value is zero, and it increases as homogeneity decreases. Low HI values indicate a good homogeneity in the dose distribution within the target volume. 

Being based on a *D_max_*/*D_min_* ratio—where *D_max_* and *D_min_* represent the maximum and minimum point dose in the target volume, respectively—the HI value is sensitive to some dose calculation parameters, such as the grid size and placement, as well as the dose gradient. As the doses can be very high or very low if only considered at a point of the dose distribution, minimum or maximum dose values are not always the best indicator. For this reason, maximum or minimum doses are rather selected in a volume (D5%, D95%, D2%, and D98% etc.) of the dose distribution.

In our study, we chose to use a classic clinical homogeneity index, HI98 [26], which was selected for homogeneity evaluation. HI98 is calculated as follows:HI98 = (*D*_2_−*D*_98_)/*D*_50_,(1)
where *D*_2_, *D*_98_, and *D*_50_ are the minimal doses delivered at 2%, 98%, and 50%, respectively, along the beam axis. In clinical practice, an HI98 value equal or less than 0.12 is considered acceptable [27].

For the mouse, the target was taken as the area corresponding to the brain (from 0.2 cm to 1.2 cm along the beam axis), the lungs (from 0.6 cm to 1.8 cm along the beam axis), and the intestine, considering a lateral beam (from 0.2 cm to 2.5 cm along the beam axis) and an anterior beam (from 0.1 cm to 1.3 cm along the beam axis). 

For biological samples, the target was taken as the two first centimeters of the plateau. In case of high homogeneity, HI98 is close to zero.

Given the size of the computing grid (0.2 mm), variations of the Bragg peak position inferior to 0.2 mm are considered null. For dose differences in the plateau, these variations are considered null when the uncertainty is inferior or equal to statistical uncertainty (±1%). Similarly, variations of HI98 inferior to ±0.02 are considered as non-significant.

## 3. Results

The results of each uncertainty study are presented in Table 3.

### 3.1. Beam Energy Spread

As shown in Figure 4, nominal spreads of ±0.68 MeV and ±1.00 MeV decreased the peak-to-plateau dose ratio and broadened the Bragg peak, compared to our reference of mono-energetic beam. However, no impact was observed on the dose delivered in the plateau, the HI98, and the proton range, whatever the beam energy spread. The position of the Bragg peak was 37.1 mm without the scatterer and 36.3 mm with the scatterer.

### 3.2. Beam Components

A variation of ±5% in the beam component thicknesses had no impact on the Bragg peak range, but a maximal impact on the absorbed dose in the plateau of ±3.5%. This was almost exclusively due to the presence of tungsten. Proton scattering increased with the thickness of the tungsten foil, in contrast to the absorbed dose in the target. 

Similarly, a variation of ±1 cm in the position of the beam components had no impact on the Bragg peak range, but a maximal impact on the absorbed dose in the plateau of ±4.7%, without tungsten. Equivalent uncertainty with tungsten was ±2.7%. The absorbed dose increased when the collimator target distance decreased.

### 3.3. I-Values of the Beam Components

The variation in *I*-values (Table 2) had no significant impact on the position of the Bragg peak: it was inferior to 0.2 mm, which is not meaningful with respect to our significance threshold. More generally, a rise in *I*-value increased the depth of the Bragg peak and vice versa. The impact on the absorbed dose was also insubstantial in the plateau. In any case, HI98 was never significantly affected.

### 3.4. Target Position

An offset of ±1 cm in the positioning of the biological sample or the mouse in the beam axis resulted in a maximal variation of ±4.4% of the absorbed dose in the plateau, for the beam scattered with the tungsten foil (Table 3). This variation was reduced to ±1.4% in the absence of tungsten. 

We also observed that the absorbed dose in the plateau increased when the collimator target distance decreased. However, this dose variation may be higher when the target is very close to the collimator. In this study, we kept a minimum distance of 4 cm between the collimator and the target entrance. For the mouse, this variation depended of the target localization and was maximal for the brain. 

The impact on the absorbed dose decreased linearly with the offset, and became insignificant below ±4.5 mm. Globally, HI98 slightly increased with the source target distance and vice versa. These variations were under the significance threshold, except for the brain, for an offset of +1 cm, where HI98 was increased from 0.09 to 0.11. The proton range was not significantly affected, and the Bragg peak position was maintained outside the target. 

### 3.5. CT Imaging and Calibration

CBCT noise in the range of ±40 HU slightly modified the proton range (by ±0.3 mm). The proton range decreased when the HU increased and vice versa. In contrast, CBCT noise had no significant impact on the absorbed dose in the plateau or the HI98 along the beam axis.

### 3.6. CT Grid Size

For the three studied localizations, the CT grid size had no significant impact on the absorbed dose in the plateau. However, the proton range was significantly impacted, with a maximal deviation of ±0.6 mm for the worst spatial resolution (0.8 × 0.8 × 0.8 mm^3^). The HI98 along the beam axis was not significantly impacted by the degradation of the spatial resolution, excepted from the lungs, for which the HI98 value increased from 0.12 to 0.14, using a spatial resolution of 0.8 × 0.8 × 0.8 mm^3^. Variations of the HI98 and the Bragg peak position observed for an intermediate spatial resolution of 0.4 × 0.4 × 0.4 mm^3^ were below the significance thresholds.

### 3.7. I-Values in Water and Tissues

A variation of the biological tissue *I*-values (±6%) and the water target *I*-value (78 ± 2 eV) had no significant impact on the absorbed dose in the plateau and the homogeneity in the target along the beam axis, but led to a maximal offset of ±0.2 mm of the proton range.

## 4. Discussion

In this study, we used Monte Carlo simulations to investigate the impact of various sources of uncertainty on the proton range, the absorbed dose, and the dose homogeneity, for pre-clinical irradiations of cells or small animals. A significant impact on the proton range was observed when varying the sources of uncertainty dependent on the dose calculation. In contrast, the absorbed dose was impacted by sources of uncertainty independent on the dose calculation. Overall, the highest impact was observed for mouse with tungsten foil, with a variation in the proton range of ±0.7 mm, and a maximal variation of ±6.5% in the absorbed dose. The presence of tungsten on the beamline led to an increase in the absorbed dose variation. However, in both cases, a commissioning measurement at each experiment allowed us to reduce this variation by even a factor of 2 in the absence of tungsten (4.8% with tungsten against 2.4% without). 

Regarding the proton range, we found that it was not impacted by the thickness or the position of the beam components. For CT imaging and calibration, CT grid size, and *I*-values in tissues, the proton range uncertainties were ±0.3 mm, ±0.6 mm, and ±0.2 mm, respectively. This corresponds to 0.8%, 1.7%, and 0.6%, respectively, for an initial range of 36 mm, which is consistent with Paganetti’s results (±0.5%, ±0.3%, and ±1.5% of initial range, respectively) [6]. As expected, we found a higher impact of CT imaging and calibration due to a higher noise level in the CBCT used in our study. The use of a µCT for small animal imaging instead of a µCBCT could provide a level of uncertainty similar to the clinical level. The impact of CT grid size is significantly higher in a preclinical situation, due to the small size and the heterogeneity of the mouse anatomy, compared with swine’s lungs used by Paganetti [6,28]. On the contrary, the impact of *I*-values for tissues is higher for the clinical study, which considers a range of *I*-values on the order of 10–15% [29]. We neglected the uncertainty values on the proton range due to complex and local lateral inhomogeneities estimated at ±0.1% by Paganetti [6], because these values are very low (0.036 mm for an initial path of 36 mm), and lower than our sensitivity threshold set at 0.2 mm. 

The proton range was significantly impacted by the sources of uncertainty related to the dose calculation. This impact stayed low (±0.2 mm) for the biological samples, but reached ±0.7 mm for mice, mainly due to the extreme CBCT grid size degradation of 0.8 × 0.8 × 0.8 mm^3^. Moreover, this grid size significantly degraded the HI98 along the beam axis. Using a maximal CBCT grid size of 0.4 × 0.4 × 0.4 mm^3^, the proton range uncertainty could be reduced to ±0.4 mm instead of ±0.7 mm and the HI98 along the beam axis could be preserved. The possible variation of the energy spread of the beam had no impact for an irradiation in the plateau. However, we observed a major impact of the peak height and width (Figure 4). This point will be critical for irradiation in the Bragg peak or in the spread-out Bragg peak. An additional study will have to be carried out to estimate the uncertainty in this context [4].

Regarding the absorbed dose, the major factors that affect the dose delivered in the Bragg curve are the uncertainties related to the beam elements, in particular their thicknesses and positions. The thickness is fixed, whereas the position can possibly vary from one time to another. Here, we chose to study the impact of positioning within ±1 cm. We observed an impact of the presence or lack thereof of the tungsten foil. Indeed, a variation in thickness of ±5% of the beam elements had no significant impact on the dose in the plateau, except in the presence of the tungsten foil (±3.5%). On the contrary, the presence of the tungsten foil reduces the impact of the position of the beam elements on the absorbed dose in the plateau. This is practically doubled (±4.7%) in the absence of tungsten. Thus, when the impact of both thickness and position are cumulated, the presence of tungsten does not significantly modify the global uncertainty on the dose in the plateau related to the beam. 

In any case, if a reference measurement is carried out at each new set-up, before installing the target, the position and the thickness of the beam components are taken into account in the measurement. These uncertainties may therefore not be included in the total uncertainty calculation. The total uncertainty of an absorbed dose in the plateau is then reduced to ±4.8%. However, the target position cannot be considered in the commissioning measurement, and its impact is not negligible. Particular care must be taken in positioning the target. Overall, our results show that the source target distance must be controlled to ±4.5 mm, in order to limit the uncertainty on the delivered dose to ±1%. Moreover, a minimal distance of several centimeters must be guaranteed between the target surface and the collimator. 

Lastly, we studied the impact of sources of uncertainty on the dose homogeneity. This is of particular interest for projects related to the limitation of toxicity to normal cells during proton beam irradiation [30,31]. The homogeneity of the target, characterized by the HI98 value, can be difficult at a low energy (68 MeV)—therefore, a short length of the plateau. In the literature, the maximum recommended clinical value of HI98 corresponds to a range of +7% and −5% of the prescribed dose [27]. A tungsten-free configuration allows a water target as thick as 20 mm to be treated with a satisfactory HI98. This thickness can be reduced to 15 mm with a tungsten foil. For small animals, if we consider a lateral beam, the HI98 calculated for the brain and lungs are 0.09 and 0.12, respectively, along the beam axis. These values are comparable to the recommended maximal clinical value of 0.12 [27]. However, this value can reach 0.31 for the intestinal target, due to the large thickness of the abdomen. In this case, it is preferable to irradiate with an anterior beam, which reduces the HI98 to 0.13 (Figure 1). Consequently, using two opposite beams could drastically improve the homogeneity of the dose in the target [32]. More generally, when feasible, the use of several beams with various angles could also reduce the impact of proton range uncertainty.

## 5. Conclusions

In this study, we estimated the impact of various sources of uncertainty specific to our proton beam line, as recommended by the American Association of Physics in Medicine (AAPM) report 202 [4]. To that aim, Monte Carlo simulations were computed to evaluate the impact of uncertainties on proton range, absorbed dose, and dose homogeneity along the beam axis in the context of preclinical irradiation of either cells or small animals irradiated in the Bragg curve. Each source of uncertainty was considered independently. For any configuration of the irradiation line, with the presence or not of a scatterer, we found that uncertainties on the proton path ensure a Bragg peak outside the target. For mouse irradiation, we recommend that dose calculations should be performed on µCT or µCBCT, with an adequate spatial resolution according to the small size of animals, in order to limit the proton range uncertainty to ±0.4 mm and to preserve dose homogeneity. Overall, uncertainties on the dose in the Bragg curve are significant, but can be reasonably reduced to ±4.8% by a systematic commissioning measurement. At the very least, a positioning accuracy better than 4.5 mm leads to a ±2.2% decrease in the global uncertainty. The evaluated uncertainties, comparable to clinical practice, appear acceptable to conduct radiobiological experiments with the ARRONAX proton beam. More generally, it appears possible with the use of Monte Carlo simulations to build a preclinical line in structures that have not been specifically designed for this application.

## Figures and Tables

**Figure 1 cancers-13-01889-f001:**
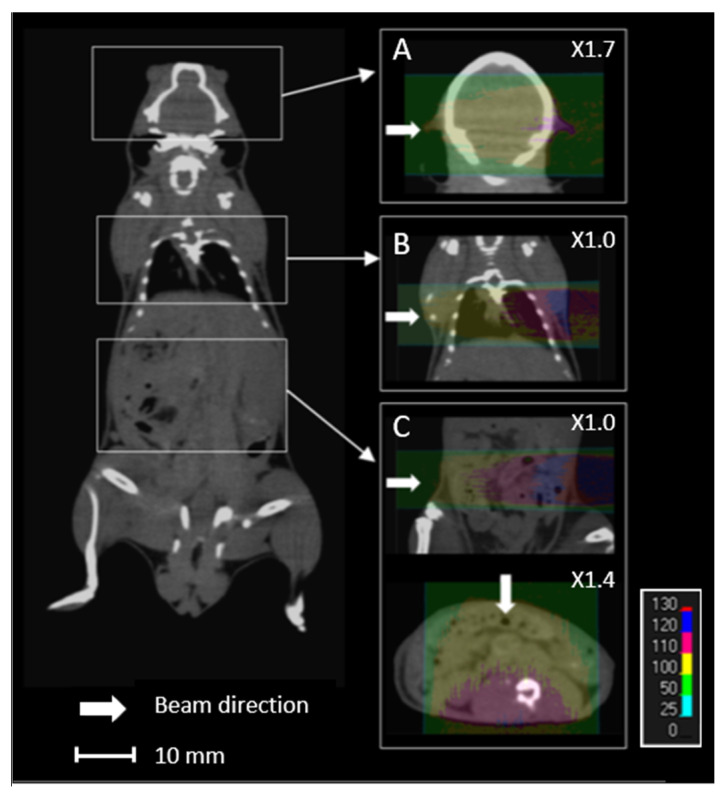
Overview of an overlay of a mouse cone beam computed tomography (CBCT) scan and the relative dose (%) obtained from simulated data for 68 MeV proton beam irradiation. Each line (**A**–**C**) comprises the simulated data of the brain, lungs, and intestines, respectively, with lateral (**A**–**C**) and anterior (**C**) proton beams.

**Figure 2 cancers-13-01889-f002:**
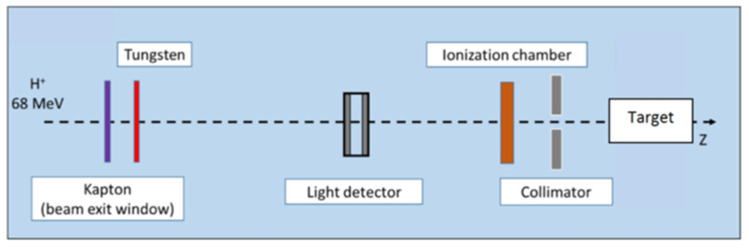
Proton beam setup. In this configuration, tungsten foil is used to homogenize the beam, for beam diameters greater than or equal to 1 cm. The target is a water tank or a mouse. Positions of each component are given in Table 1.

**Figure 3 cancers-13-01889-f003:**
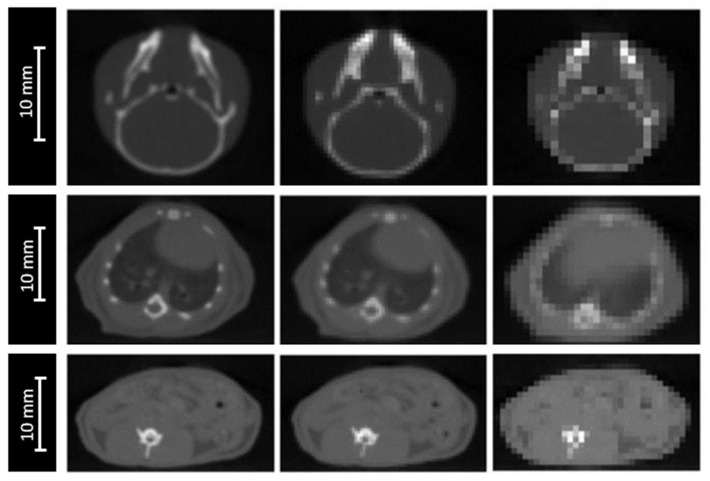
CBCT of a mouse’s head (top), thorax (middle). and abdomen (bottom) for three different CBCT grid sizes: 0.2 × 0.2 × 0.2 mm^3^ (left), 0.4 × 0.4 × 0.4 mm^3^ (middle), and 0.8 × 0.8 × 0.8 mm^3^ (right).

**Figure 4 cancers-13-01889-f004:**
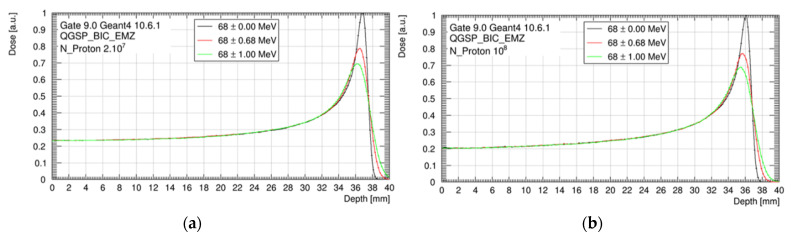
Bragg peak for three beam energy configurations (68 ± 0.00 MeV; 68 ± 0.68 MeV and 68 ± 1.00 MeV), considering the setup without tungsten foil (**a**) and with tungsten foil (**b**). Data are normalized to the maximum of the monenergetic Bragg peak curve.

**Table 1 cancers-13-01889-t001:** Material, position, thickness, and shape of the beam line components used in GATE modeling.

Components	Shape	Material	Position (cm)	Thickness (cm)
Inside of the beam pipe	Cylinder	Vacuum	−5.00	10.00
Kapton window	Cylinder	Kapton	−2.50 × 10^−3^	5.00 × 10^−3^
Tungsten foil (optional)	Box	Tungsten	1.20	5.17 × 10^−3^
Light detector	Box	Aluminum	2.01	2.00 × 10^−3^
		Air	4.51	5.00
		Aluminum	7.02	2.00 × 10^−3^
Ionization chamber	Box	Copper	161.00	2.00 × 10^−4^
Final collimator	Hollow cylinder	Aluminum	163.75	2.50
Water phantom (biological sample)	Cylinder	Water	169.50	4.00
Mouse	CT	125 tissues	171.00	(1.50–2.70 cm)

**Table 2 cancers-13-01889-t002:** *I*-values and corresponding uncertainties applied to the beam components.

Material	*I*-Value (eV)	Uncertainty (eV)
Aluminum (Al)	166.0 *	±2.0 *
Tungsten (W)	727.0 *	±30.0 *
Kapton (C_22_H_10_N_2_O_5_)	79.6 *	±4.8 **
Air	85.7 *	±5.1 **

* International Commission on Radiation Units and Measurements (ICRU) report 37 [14]; ** Bär et al. [16].

**Table 3 cancers-13-01889-t003:** Estimated dose uncertainties in the Bragg curve and proton range uncertainties, using Monte Carlo dose simulation. NS = not significant, considered as null, if uncertainty <±0.2 mm (calculation grid size) or <±1.0%. Total uncertainties are equal to the square root of the sum of squares of considered uncertainties.

Source of Uncertainty	Range Uncertainty (mm)	Dose Uncertainty in the Bragg Curve (%)	
Independent of Dose Calculation:		With scatterer (tungsten foil)	Without scatterer
Measurement uncertainty for commissioning	-	±2.0%	±2.0%
Beam energy spread * (spread ±1 MeV)	NS	NS	NS
Beam component thickness (±5%)	NS	±3.5%	NS
Beam component position * (±1 cm)	NS	±2.7%	±4.7%
Biological sample or animal setup in the beam axis * (+/1 cm)	NS	±4.4%	±1.4%
Dependent on dose calculation:			
CT imaging and calibration (±40 HU) *^	±0.3 mm	NS	
CT grid size (0.2/0.4/0.8 mm) ^	±0.6 mm	NS	
Mean excitation energy (*I*-values) of beam components	NS	NS	
Mean excitation energy (*I*-values ±6%) in tissues ^	±0.2 mm	NS	
Mean excitation energy (*I*-values) in water +	±0.2 mm	NS	
Overall			
For biological samples (excluding ^)	±0.2 mm	±6.5% (±4.8% **)	±5.3% (±2.4% **)
For mice (excluding +)	±0.7 mm	±6.5% (±4.8% **)	±5.3% (±2.4% **)

* Can vary over time and between experiment; ** considering a commissioning measurement at each experiment. In the present case, “total uncertainties” are equal to the square root of the sum of squares of “measurement uncertainty for commissioning” and “biological sample or animal setup in the beam axis”. The symbols “^” and “+” indicate the categories excluded in the calculation of the overall part. CT = Computed Tomography; HU = Hounsfield Unit.

## Data Availability

Data available on request due to restrictions eg privacy or ethical. The data presented in this study are available on request from the corresponding author. The data are not publicly available due to their extended size, incompatible with online upload.

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
