# Peer review of "A Monte Carlo Determination of Dose and Range Uncertainties for Preclinical Studies with a Proton Beam"

_cancers, 2021, doi:10.3390/cancers13081889_

Round 1
Reviewer 1 Report
This manuscript presents a detailed assessment of uncertainties associated to the ARRONAX cyclotron hardware or to the irradiated target, and their impact on proton range and dose deposition.
Overall the manuscript is of high quality. Data presentation could be improved in some areas. Also, a lot of "reference errors" were present in the pdf version. I do not blame anyone but it was a bit tricky to read on occasions.
Specific comments:
- Section 2.2: MC simulation: please add details on the two target geometries (mice or cells ?). Is the cell geometry a monolayer ? In that case, the 100 µm step size is larger than the target which undermines the results a bit
- Section 2.4: studied parameters are proton range and dose in the plateau. Why would you study the impact on proton range for shoot through configurations ? I understand it is useful data for future studies (e.g. irradiation of mice in SOBP), but it sounds a bit out of scope. Still in this section, please expand a bit on HI98 definition. The target audience might not be familiar with the concept. Still in this section, the biological samples (different from the mouse case ?) thickness is reported to be 2 cm. If it refers to cells, again it is 4 orders of magnitude off. Please clarify.
- Table 3: It became more or less clear in the text by reading through, but the global error is calculated either by including measurement uncertainty for commissioning OR beam components thickness and position. Did I got this right ? There is probably a nice way to make it clear directly in table 3.
- A table presenting results about HI98 would help.
Author Response
Please see the attachment.
Kind regards,
Arthur Bongrand

Reviewer 2 Report
The manuscript is very well conceived and well written in adaquate English language. The main output of the work is clearly described and presented in a optimal form. The manuscript deals deeply with a ever-growing topic of interest. The ‘introduction’ paragraph is centered on the aim of the work, with a broad look at the results in the literature. The 'materials and methods' part is fully described with adequate methodologies of analysis for the investigation, using Monte Carlo simulations. The 'results’ and ‘discussion' parts explain all results clearly and step-by-step. The manuscript is readable thanks to the figures and table. The topic of the manuscript is of interest to readers of the Journal.
Few minor modification are required.
- At the end of line 45, add the following useful reference: Younkin, J.E.; Bues, M.; Sio, T.T.; Liu, W.; Ding, X.; Keole, S.R.; Stoker, J.B.; Shen, J. Multiple energy extraction reduces beam delivery time for a synchrotron-based proton spot-scanning system. Advances in Radiation Oncology 2018, 3, 412-420. doi: 10.1016/j.adro.2018.02.006.
- At the end of line 65, add the following adequate reference (that focus on Monte Carlo simulations description): Ambrosino, F.; Roca, V.; Buompane, R.; Sabbarese, S. Development and calibration of a method for direct measurement of 220Rn (thoron) activity concentration. Applied Radiation and Isotopes 2020, 166, 109310, doi:10.1016/j.apradiso.2020.109310.
- In the manuscript, the following error occurs several times: “Error! Reference 197 source not found”. Please correct.
- Line 245, breaks the sentence “usingMonte”.
- Line 257, add the same significant digits as in the figure 4.
I recommend to accept the manuscript for publication in the Journal, only after taking into account the above list of minor modifications that I required.
Author Response

(The authors gave the same response as above.)
